# Does Green Brazilian Propolis Extract Improve Functional Capacity in Symptomatic Chronic Coronary Disease?—A Pilot Randomized Trial

**DOI:** 10.3390/ph18060827

**Published:** 2025-05-31

**Authors:** Clara Salles Figueiredo, Luiz Carlos Santana Passos, Caio Rebouças Fonseca Cafezeiro, Rodrigo Morel Vieira de Melo, Tainá Teixeira Viana, Eduardo Jorge Gomes de Oliveira, Andresa Aparecida Berretta, Marcelo Augusto Duarte Silveira

**Affiliations:** 1Programa de Pós-Graduação em Medicina e Saúde (Graduate Program in Medicine and Health), Faculdade de Medicina da Bahia, Universidade Federal da Bahia, Salvador 40110-060, BA, Brazil; 2Hospital Ana Nery, Salvador 40301-155, BA, Brazil; 3D’Or Institute for Research and Education (IDOR), Hospital São Rafael, Avenida São Rafael 2152, São Marcos, Salvador 41253-190, BA, Brazil; 4Development and Innovation Department, Apis Flora Indl. Coml. Ltd., Rua Triunfo 945, Subsetor Sul 3, Ribeirão Preto 14020-670, SP, Brazil

**Keywords:** propolis, green propolis, anti-inflammatory, chronic coronary disease, coronary heart disease, angina pectoris, EPP-AF

## Abstract

**Background:** Inflammation plays a critical role in the progression of coronary heart disease (CHD). Low-dose colchicine has shown promise in reducing cardiovascular events, and green Brazilian propolis extract (EPP-AF® (standardized Brazilian green propolis extract) was provided by Apis Flora Indl. Coml. Ltda, Ribeirão Preto, SP, Brazil), known for its anti-inflammatory properties, may offer additional therapeutic benefits. This pilot study aimed to evaluate whether six weeks of EPP-AF^®^ supplementation improves functional capacity assessed by treadmill exercise testing. **Methods**: This was a randomized, double-blind, placebo-controlled pilot study conducted at a coronary disease clinic in Brazil. Patients aged ≥ 18 years with stable CHD receiving optimized medical therapy were randomized in a 2:1 ratio to receive either 200 mg of EPP-AF^®^ or placebo twice daily for six weeks. The primary outcome was the change in treadmill exercise duration (in seconds). Secondary outcomes included total exercise time, functional capacity (measured in metabolic equivalents of task [METs]), high-sensitivity C-reactive protein (hs-CRP) levels, the Seattle Angina Questionnaire (SAQ), and the Canadian Cardiovascular Society (CCS) angina classification. Statistical analysis was performed on an intention-to-treat basis. **Results**: A total of 59 patients were randomized, with a median follow-up of 6.5 weeks. There was no significant difference in the primary endpoint between groups: the median change in treadmill test time was 39 s in the EPP-AF^®^ group versus 30 s in the placebo group (*p* = 0.83). No improvements were observed in METs, hs-CRP levels, SAQ scores, or CCS class in the EPP-AF^®^ group. No major adverse cardiovascular events occurred during the study. **Conclusions:** EPP-AF^®^ did not improve functional capacity, inflammatory markers, or angina symptoms in patients with stable CHD compared to placebo.

## 1. Introduction

Coronary heart disease (CHD) remains the primary global cause of death. Its chronic manifestation as stable angina is associated with reduced quality of life and significant disability-adjusted life years [1,2]. Optimal medical therapy (OMT), according to current guidelines for chronic coronary disease (CCD), involves administration of lipid-lowering medications and antiplatelet agents to slow the progression of atherosclerosis, along with anti-anginal medications for symptom management [1,3].

In recent years, inflammation has been recognized as a key contributor to atherogenesis and the pathophysiology of ischemic events. Consequently, it has emerged as a promising therapeutic target, as demonstrated by numerous recent clinical trials [4,5]. The CANTOS trial was the first to test the hypothesis that targeting inflammation could improve cardiovascular outcomes in patients with CHD [6], using canakinumab, an interleukin-1β inhibitor. Building on this concept, subsequent trials investigated other anti-inflammatory strategies, notably demonstrating that low-dose colchicine can reduce the incidence of myocardial infarction, stroke, and revascularization due to unstable angina [7,8]. These findings ultimately led to the recent indication of low-dose colchicine for secondary prevention in this population [3].

Standardized Brazilian green propolis extract (EPP-AF^®^) is a natural resinous substance collected by bees from plant exudates and is rich in bioactive compounds such as flavonoids, phenolic acids, and terpenes [9]. These components have demonstrated potent anti-inflammatory, antioxidant, and immunomodulatory properties in both in vitro and in vivo studies [10,11]. The cardiovascular effects of EPP-AF^®^ are not yet fully understood, but some existing evidence suggests its potential cardioprotective role through mechanisms such as modulation of lipid metabolism, attenuation of oxidative stress, and protection against endothelial dysfunction [12]. Thus, EPP-AF^®^ emerges as a promising natural anti-inflammatory agent with potential benefits in the management of CCD.

The aim of this study was to conduct a randomized, controlled, double-blind pilot trial to assess whether the use of EPP-AF for six weeks in symptomatic patients with CCD and OMT would be associated with an improvement in functional capacity, as demonstrated by treadmill tests.

## 2. Results

A total of 64 patients were initially enrolled in the study, but five did not meet the primary inclusion criteria. Consequently, 59 patients were randomized and received at least one dose of either propolis or placebo. Five participants were lost to follow-up. In addition, one patient declined to undergo the baseline and follow-up exercise tests, and two patients could not complete the final treadmill test due to mobility issues after six weeks of treatment (Figure 1).

The baseline characteristics of the patients were well balanced between the trial groups (Table 1). The median age of patients was 61 years (57–67), and 37 (62.7%) were male; 30 (50.8%) had diabetes, 26 (54%) had previous STEMI, and 34 (57.6%) had multi-vessel coronary disease. At baseline, patients were receiving appropriate treatment for CAD, with a median LDL-c of 79 mg/dL (60–105), and 56 (95%) were on aspirin. Moreover, 33 (56%) of patients reported angina symptoms classified as CCS grade 2 or 3, despite the use of multiple anti-anginal medications, including beta-blockers (57, 96.6%), CCB (42, 71%), nitrates (27, 46%), and trimetazidine (25, 42.3%).

The median follow-up duration was 6.5 weeks (5.8–7.5 weeks). Two patients permanently discontinued the intervention due to adverse events related to gastric intolerance.

### Outcomes

The primary endpoint, defined as the median treadmill test time from baseline to week 6, showed an increase of 39 s (IQR: −59 to 128) in the propolis group and an increase of 30 s (IQR: −10 to 101) in the placebo group (*p* = 0.83) (Figure 2). The treadmill test duration showed a numerical improvement in both groups, with no difference in the propolis group, increasing from 367 s (IQR: 273–555) to 378 s (IQR: 232–558) (*p* = 0.47). Functional capacity, measured in METs, also increased numerically in the second exercise test for both groups, though this change was not statistically significant compared to baseline (Table 2). Additionally, there were no significant differences in hs-CRP levels between baseline and the follow-up period in either the propolis or placebo groups (Table 2).

At the end of the follow-up period, 10 (31%) of patients in the propolis group and 10 (41%) in the placebo group were classified as having CCS grade 2 or 3 angina, with no statistically significant differences from baseline (*p* = 0.21 for propolis and *p* = 0.06 for placebo). No changes were observed in the Seattle Angina Questionnaire 7-item (SAQ-7) scores across all domains for patients receiving propolis over the six-week follow-up (Table 3).

## 3. Discussion

The PRAIA trial did not show that standardized Brazilian green propolis extract improved functional capacity in symptomatic patients with stable coronary artery disease already on guideline-based medical therapy. Specifically, there was no improvement in treadmill exercise duration compared to placebo. Additionally, this intervention was not associated with improvements in other outcomes, such as total exercise duration, functional capacity measured in METs, reductions in hs-CRP levels, or symptom relief as assessed by the CCS score and SAQ-7.

Inflammation plays a central role in atherosclerosis, contributing to endothelial dysfunction and plaque rupture [5,13,14]. Recent studies have demonstrated the benefits of targeting inflammation to reduce major adverse cardiac events in CAD patients [7,15,16]. Additionally, elevated inflammatory markers have been shown to correlate with the severity and prognosis of stable angina [14,17].

The main goal in treating chronic coronary syndrome is to improve quality of life and life expectancy [3]. However, the range of available treatments remains limited, highlighting the need for new alternatives. Recently, low-dose colchicine, an established anti-inflammatory drug, has shown significant cardiovascular benefits in some randomized, placebo-controlled trials. It has been linked to fewer hospitalizations for worsening angina and is now included in guidelines for managing chronic coronary disease [15].

Propolis is a natural product produced by bees from a combination of plant resins and bioactive compounds. It is a complex mixture containing phenolics, flavonoids, aromatic compounds, terpenes, alcohols, β-steroids, and aldehydes [12,18]. Propolis has long been recognized for its anti-inflammatory, antioxidant, and immunomodulatory properties [9]. In a meta-analysis including 406 individuals, Jalali et al. demonstrated that propolis use was associated with significant reductions in inflammatory markers such as tumor necrosis factor-alpha (TNF-α) and C-reactive protein (CRP) [19].

The cardiovascular effects of propolis have been primarily studied in vitro experiments and animal models [12]. Available evidence suggests that propolis acts as an anti-inflammatory agent by preventing vascular endothelial dysfunction and reducing inflammatory markers [9]. These promising effects underscore the potential of EPP-AF^®^ as a therapeutic agent for cardiovascular diseases. A recent study with EPP-AF^®^ in patients with high cardiovascular risk on hemodialysis showed a significant reduction in hs-CRP [20]. This is the first clinical trial to evaluate propolis as an anti-inflammatory and anti-anginal therapy in patients with coronary heart disease.

While the PRAIA trial did not demonstrate the efficacy of propolis, interest in alternative therapies for cardiovascular diseases continues to grow. Several natural compounds with anti-inflammatory properties have shown potential in preclinical and clinical settings [4]. For instance, traditional Chinese medicine (TCM) formulations, such as Tongxinluo and Guanxin Danshen dropping pills, have been evaluated in randomized trials for angina and coronary artery disease [21]. Although these interventions differ in composition and context, they share with EPP-AF^®^ a proposed anti-inflammatory mechanism in CHD—a therapeutic approach that continues to be explored.

Similarly, *Terminalia arjuna*, a plant-based compound with antioxidant and anti-inflammatory properties, has been studied in patients with coronary artery disease, showing reductions in inflammatory cytokines, though without significant effects on clinical outcomes [22]. These examples highlight the broader scientific interest in bioactive natural products for cardiovascular health and support further investigation of agents like EPP-AF^®^ in this field.

Although our results were negative, it is important to note that both groups showed an improvement in total exercise time at the end of the follow-up period. This effect may, in part, be attributed to the Hawthorne phenomenon, in which patients modify their behavior simply due to the awareness of being observed. Ischemic preconditioning may also have contributed, as repeated brief episodes of ischemia can induce protective adaptations that enhance myocardial tolerance to subsequent exertion. Additionally, regression to the mean—a statistical phenomenon common in studies enrolling symptomatic patients during periods of exacerbation—may explain part of the apparent improvement, particularly given the short study duration.

Another possible explanation for the neutral findings is that only approximately one-third of participants had baseline hs-CRP levels >2 mg/L, suggesting that many patients did not present with active systemic inflammation. As such, the study may not have adequately tested the efficacy of targeting inflammatory pathways in CHD to reduce cardiovascular events. Prior trials, such as CANTOS, demonstrated that patients with elevated hs-CRP who achieved reductions below 2 mg/L experienced greater clinical benefits [6]. Similarly, the lack of effect in the recent CLEAR-SYNERGY trial has been partly attributed to insufficient inflammatory suppression [23]. These considerations highlight the importance of selecting populations with evidence of active inflammation and suggest that future trials with propolis should include hs-CRP thresholds as part of their inclusion criteria to better assess its anti-inflammatory potential.

The PRAIA trial did not demonstrate the clinical efficacy of green Brazilian propolis in CAD; however, its significance as a randomized study evaluating potential treatments for angina should not be overlooked. There is a lack of robust evidence from large, randomized trials for the primary anti-anginal medications recommended in recent guidelines. Most of these drugs have only been tested in smaller trials comparing them to beta-blockers, with a notable gap in head-to-head studies. Therefore, this area of care demands more attention, particularly given the poor prognosis and diminished quality of life for these patients.

The study has limitations, including a small sample size, which may have limited statistical power and contributed to potential type I and type II errors. Secondly, the 6-week follow-up period, though typical for anti-anginal therapy trials [3], may have been too short to capture the long-term effects of the intervention. Future studies should address these issues by increasing sample size, extending follow-up, and using a multicenter approach to improve validity. The sample size for this pilot study was calculated based on the assumption that it would be sufficient to detect differences in functional capacity in patients with CAD based on previous anti-anginal trials [24,25]. Additionally, although the analysis followed the intention-to-treat principle, the absence of a per-protocol analysis due to participant dropout may limit the interpretability of efficacy outcomes.

In conclusion, the administration of standardized Brazilian green propolis extract (EPP-AF^®^) did not lead to significant improvements in functional capacity on the treadmill, reduction of inflammation markers tested by hsCRP, or angina symptoms compared to placebo in patients with stable coronary artery disease. However, its safety profile and known bioactive properties justify further investigation into cardiovascular diseases through larger, multicenter trials with extended follow-up periods.

## 4. Materials and Methods

### 4.1. Study Designed and Patients

The PRAIA (effect of green propolis extract in chronic stable angina patients) (a randomized, double-blind, pilot study) trial was conducted in a coronary disease clinic in an outsourced center in Salvador, Bahia, Brazil. This clinical trial was conducted based on a prespecified protocol registered in the Brazilian Registry of Clinical Trials (RBR-876w38c) and received ethical approval from the local ethical committee (registration number 36882820.9.000.0045). All data generated or analyzed during this study are included in this published article. The present study was conducted in accordance with the principles of the Good Clinical Practice guidelines of the International Conference on Harmonization. The provision of EPP-AF^®^ active drug or placebo was funded by the Collaborating Institution for the Production and Distribution of Propolis: Apis Flora Industria Comercial LTDA.

Patient recruitment took place from May 2021 to January 2023. The patients were identified consecutively during outpatient care carried out at clinics specialized in coronary artery disease at the study center. Potential participants were invited for a screening visit and underwent comprehensive assessment. All patients provided informed consent and were enrolled if all inclusion criteria and none of the exclusion criteria were met.

Patients were eligible if they were aged 18 years or older, had any evidence of coronary disease on invasive coronary angiography or computed tomography angiography (≥50% narrowing in at least one coronary artery), and experienced stable angina symptoms (or ischemic equivalents such as dyspnea or arm pain with exertion) despite optimized treatment with lipid-lowering drugs, antiplatelets, and anti-anginal medications (beta-blockers, calcium channel blockers, long- and short-acting nitrates, and/or trimetazidine) for at least 4 weeks.

Exclusion criteria included patients scheduled for myocardial revascularization within 30 days, those with left main coronary artery obstruction more than 50%, recent acute coronary syndrome (less than 3 months), baseline electrocardiogram (ECG) abnormalities that might interfere with the interpretation of ECG ST segment changes, physical limitations for treadmill testing, symptomatic NYHA III-IV heart failure with reduced ejection fraction, and participation in another research protocol.

### 4.2. Randomization and Follow-Up

Patients were randomly assigned by computer and managed by a research assistant who was not involved in the evaluation or management of study patients. Randomization was performed in a double-blind manner in a 2:1 allocation ratio to receive 200 mg twice a day of propolis extract (EPP-AF^®^) or matching placebo, with stratification according to diabetes status, angina severity by the Canadian Cardiovascular Society Classification for Angina (CCS), and number of anti-anginal drugs prescribed. The active and placebo capsules were identical in appearance, color, odor, and taste and were packaged in identical containers to ensure effective blinding.

Clinical assessments and maximal treadmill exercise tests were performed at the initial assessment and at the end of the follow-up period (6 weeks). Patients were encouraged to contact investigators by telephone with any questions or in the presence of clinical changes. All other treatments for CCD remained unchanged, and dose adjustments were prohibited during the six-week follow-up period.

### 4.3. End Points

The primary endpoint was to evaluate the median change, measured in seconds, from baseline to week 6 in exercise duration on the treadmill test. Secondary outcomes included the total exercise time achieved on the treadmill test, functional capacity calculated by METs, measurement of high-sensitivity C-reactive protein (hs-CRP) levels, the mean score on the Seattle Angina Questionnaire (SAQ) in the angina domains (SAQ-angina), as well as the total score obtained in all domains, and angina symptoms by the CCS assessment. Any unfavorable clinical outcomes, such as cardiovascular death, myocardial infarction (MI), stroke, and need for urgent revascularization, were documented during the study period.

The exercise test was performed following the Bruce protocol by a single research assistant. The intensity of anginal pain during exercise was assessed using the Borg scale. Positive exercise tests were defined by horizontal or downward depression ≥ 1 mm in the ST segment (60 ms after the J point) accompanied by anginal pain or depression ≥ 1.5 mm without anginal pain. In addition, the duration of the exercise test, total workload in metabolic equivalents (METs), and maximum depression of the ST segment were recorded. High-sensitivity C-reactive protein (hs-CRP) levels were quantified by the hospital laboratory using the VITROS Chemistry Products (Ortho Clinical Diagnostics, Raritan, NJ, USA). Operators were blinded to treatment group.

To assess angina-specific health status, the 7-item Seattle Angina Questionnaire (SAQ) was administered before randomization and at the 6-week follow-up. The SAQ is a validated tool, known for its reliability and sensitivity to clinical changes [26]. It measures angina frequency, physical limitation due to angina, angina stability, treatment satisfaction, and quality of life (QoL) over the preceding 4 weeks. These scores are averaged to derive the SAQ Summary score, representing overall disease-specific health status. SAQ scores range from 0 to 100, with higher scores indicating less frequent angina, improved function, and better QoL.

### 4.4. Statistical Analysis

The Kolmogorov–Smirnov test verified the normal distribution of continuous variables. Normally distributed variables were described by means and standard deviations, while non-symmetrically distributed ones were described by medians and 25th/75th percentiles. Categorical variables were reported as frequency and percentage. Chi-square tests compared categorical variables. Paired *t*-tests and Wilcoxon tests compared scores and domains from baseline to follow-up for parametric and non-parametric variables, respectively. Independent samples *t*-tests and Mann-Whitney tests compared score differences between intervention groups. Statistical significance was set at *p* < 0.05. Data were analyzed using SPSS version 20.0.

Given the exploratory nature of our study as a pilot investigation, we estimated a total sample size of 60 patients. The analysis was conducted on an intention-to-treat basis, including all randomized patients regardless of adherence to the treatment protocol.

## Figures and Tables

**Figure 1 pharmaceuticals-18-00827-f001:**
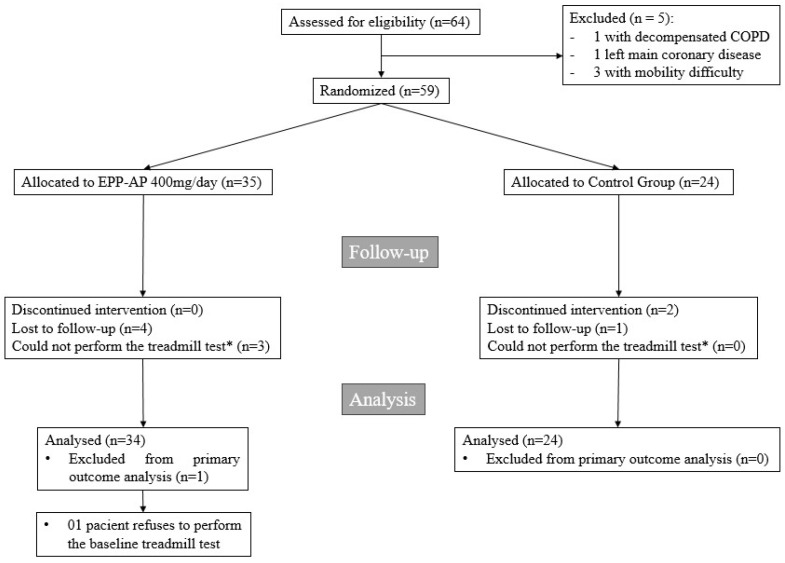
Enrollment, randomization, and follow-up. * Patients did not undergo the exercise test due to refusal or the presence of any mobility impairment.

**Figure 2 pharmaceuticals-18-00827-f002:**
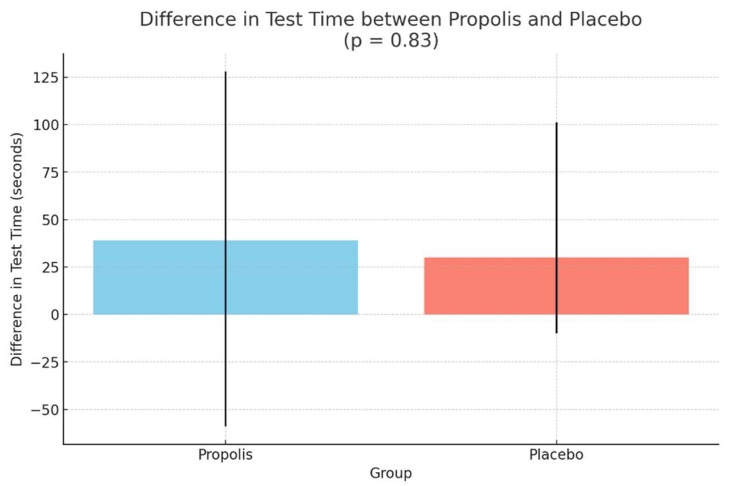
Primary end point.

**Table 1 pharmaceuticals-18-00827-t001:** Baseline characteristics.

	Propolis (N = 35)	Placebo (N = 24)
Age (IQR)	59 (53–65)	64 (59–71)
Male (%)	68.6%	54.2%
Body mass index (IQR)	27.3 (25.3–28.3)	26.7 (26–28.6)
SBP (IQR)	130 (112–154)	131 (120–150)
Heart rate (IQR)	71 (64–77)	65 (62–69)
Total cholesterol (IQR)	140 (93–171)	132 (114–181)
LDLc (IQR)	73 (45–104)	76 (58–90)
Creatinine clearance (IQR)	87 (75–101)	86 (60–98)
Hypertension (n)	31 (88.6%)	22 (91.7%)
Diabetes (n)	17 (48.6%)	13 (54.2%)
LVEF (IQR)	63 (53–67)	66 (55–74)
LVEDd (IQR)	50 (46–53)	47 (44–51)
LVESd (IQR)	31 (31–37)	31 (26–33)
STEMI (n)	15 (42.9%)	11 (45.8%)
PCI (n)	14 (40%)	8 (33.3%)
CABG surgery (n)	1 (2.9%)	0
ADA disease (n)	30 (85.7%)	20 (83.3%)
CAD multiarterial (n)	20 (57.1%)	14 (58.3%)
Angina symptoms (n) - CCS grade 1 - CCS grade 2 - CCS grade 3 - Atypical angina	13 (37.1%) 11 (31.4%) 7 (20%) 4 (4.2%)	8 (33.3%) 8 (33.3%) 7 (29.2%) 1 (11.4%)
Aspirin (n)	34 (97.1%)	22 (91.7%)
Clopidogrel (n)	10 (28.6%)	8 (33.3%)
Statins (n)	34 (97.1%)	24 (100%)
Ezetimibe (n)	12 (34.3%)	7 (29.2%)
IECA/ARB (n)	33 (94.3%)	19 (79.2%)
Anti-anginal medications - Nitrate (n) - Beta-blockers (n) - CCB (n) - Trimetazidine (n)	13 (37.1%) 34 (97.1%) 25 (71.4%) 14 (40%)	14 (58.3%) 23 (95.8%) 17 (70.8%) 11 (45.8%)
hs-CRP (IQR)	1.43 (0.38–2.90)	1.22 (0.49–3.62)
hs-CRP ≥ 2mg/L	9 (25%)	10 (42%)

CABG—coronary arterial bypass graft; CCB—calcium channel blockers; IQR—interquartile range; hs-CRP—high-sensitivity C reactive protein; LDLc—low-density level cholesterol; LVEDd—left ventricular end-diastolic diameter; LVESd—left ventricular end-sistolic diameter; LVEF—left ventricular ejection fraction; PCI—percutaneous coronary intervention; SBP—systolic blood pressure.

**Table 2 pharmaceuticals-18-00827-t002:** Secondary end points after randomization.

	**Propolis**			**Placebo**		
	**Before Treatment** **(n = 34)**	**After Treatment** **(n = 31)**	** *p* **	**Before Treatment** **(n = 24)**	**After Treatment** **(n = 23)**	** *p* **
Duration on the treadmill test (seconds) (IQR)	367 (273–555)	378 (232–558)	0.24	341 (217–380)	355 (232–443)	0.03
Functional capacity in METs (IQR)	6.70 (4.93–9.80)	7.16 (4.98–9.98)	0.36	5.98 (3.94–7.16)	6.36 (4.31–7.75)	0.06
	Propolis			Placebo		
	Before Treatment (n = 35)	After Treatment (n = 31)	*p*	Before Treatment (n = 24)	After Treatment (n = 23)	*p*
hsCRP (IQR)	0.92 (0.44–2.45)	1.14 (0.34–3.99)	0.65	1.53 (0.735–4.40)	1.24 (0.68–5.59)	0.42

IQR—interquartile range; hs-CRP—high-sensitivity C reactive protein; METs—metabolic equivalents.

**Table 3 pharmaceuticals-18-00827-t003:** Effect of propolis and placebo on the domain scores of the Seattle Angina Questionnaire (SAQ).

	Propolis			Placebo		
	Before Treatment (n = 35)	After Treatment (n = 31)	*p*	Before Treatment (n = 24)	After Treatment (n = 23)	*p*
SAQ-7 Angina Frequency Score, Median (IQR)	80 (70–85)	90 (80–100)	0.43	80 (60–80)	90 (80–100)	0.002
SAQ-7 Physical Limitation Score, Median (IQR)	55.5 (41.6–88.8)	62.5 (40.3–87.5)	0.84	66.6 (41.6–85.1)	69.4 (55.5–91.6)	0.36
SAQ-7 Quality-of-Life Score, Median (IQR)	58.3 (41.7–66.7)	50 (33.3–66.7)	0.52	50 (41.6–70.8)	66.6 (50–75)	0.055
SAQ-7 Angina Stability Score, Median (IQR)	75 (50–87.5)	75 (50–100)	0.77	50 (50–100)	100 (75–100)	0.015
SAQ-7 Treatment Satisfaction Score, Median (IQR)	93.7 (78.1–100)	93.7 (78.1–100)	0.73	87.5 (75–100)	100 (87.5–100)	0.162
SAQ-7 Summary Score, Median (IQR)	61.3 (47.4–75.3)	62.7 (50.2–75.2)	0.38	63.7 (50.2–70.5)	76.3 (59.5–84.2)	0.001

## Data Availability

The original contributions presented in this study are included in the article. Further inquiries can be directed to the corresponding author.

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
