# Peer review of "Does Green Brazilian Propolis Extract Improve Functional Capacity in Symptomatic Chronic Coronary Disease?—A Pilot Randomized Trial"

_pharmaceuticals, 2025, doi:10.3390/ph18060827_

Round 1
Reviewer 1 Report
Comments and Suggestions for Authors
The manuscript needs a major revision.
Main notes:
- After looking at the iThenticate report, I suppose that the percent match (28%) is not appropriate. Therefore, it should be corrected.
- The Introduction section is very short and should be expanded by reviewing current scientific publications. For instance, this link could be useful –
https://pubmed.ncbi.nlm.nih.gov/?term=Coronary+heart+Disease+and+propolis&sort=date&size=50
- The authors should add a separate paragraph to the discussion about the chemical composition of propolis and its bioactive components, which could exert the identified therapeutic effects.
- Why do the authors mention colchicine in the abstract and in the introduction, which is a well-known alkaloid with poisonous properties?
- The keyword list should be reviewed and improved.
- This abbreviation should be deciphered in the Abstract, as well as other terms.
- Research design should be more clearly described.
- The Discussion section should be corrected for ease of understanding. The authors should also pay more attention to comparisons with similar clinical studies of propolis –
https://clinicaltrials.gov/search?cond=Coronary%20Disease&term=propolis
https://clinicaltrials.gov/search?cond=cardiovascular%20health&term=propolis
Author Response
Author's Reply to the Review Report (Reviewer 1)
- After looking at the iThenticate report, I suppose that the percent match (28%) is not appropriate. Therefore, it should be corrected.
Thank you for your comment regarding the iThenticate report. We would like to clarify that the manuscript is entirely original and was developed based on our own research. As is common in academic writing, we used references from previously published studies to support our rationale and to discuss our findings in the context of existing literature.
We are committed to maintaining the integrity of scientific writing and would greatly appreciate it if you could kindly indicate which sections of the manuscript, you believe require attention, based on the similarity report. This will help us better understand your concerns and make the necessary adjustments, if applicable.
- The Introduction section is very short and should be expanded by reviewing current scientific publications.
Thank you for your comments regarding the introduction. I have taken your observation into consideration and added further information to enrich the text and provide a more comprehensive context.
- The authors should add a separate paragraph to the discussion about the chemical composition of propolis and its bioactive components, which could exert the identified therapeutic effects.
Thank you for the suggestion. I have added additional information regarding the chemical composition of propolis to enhance the contextual background of the manuscript.
- Why do the authors mention colchicine in the abstract and in the introduction, which is a well-known alkaloid with poisonous properties?
Thank you for your question. Colchicine was mentioned in several parts of my study because it is currently the most extensively studied anti-inflammatory agent in the context of both stable and acute coronary syndromes. In recent years, multiple large randomized clinical trials and meta-analyses have consistently demonstrated that low-dose colchicine reduces cardiovascular events in patients with coronary heart disease.
Despite its alkaloid nature and potential toxicity at higher doses, colchicine has shown a favorable safety profile at low doses and has achieved significant clinical impact—leading to its recent approval by the FDA for secondary prevention in cardiovascular disease.
We used colchicine as a reference in our article to illustrate how targeting inflammation can yield cardiovascular benefits, especially considering that other anti-inflammatory agents previously investigated (such as corticosteroids and methotrexate) have not demonstrated similar efficacy.
Therefore, the mention of colchicine helps contextualize the rationale for exploring novel anti-inflammatory strategies—such as standardized Brazilian green propolis extract—in patients with coronary artery disease.
- The keyword list should be reviewed and improved.
Thank you for the suggestion. I have reviewed and updated the keyword list accordingly.
- This abbreviation should be deciphered in the Abstract, as well as other terms.
Thank you for your observation. I have made the necessary corrections to the abbreviations in the abstract, ensuring that all terms are properly introduced and consistent throughout the text.
- Research design should be more clearly described.
Thank you for your comment regarding the description of the research design. I have included all relevant information following clinical trial reporting standards, as outlined in the CONSORT checklist provided in the supplementary material. However, I would greatly appreciate it if you could indicate which specific aspects you found unclear, so that I can revise the section accordingly and provide further clarification.
The Discussion section should be corrected for ease of understanding. The authors should also pay more attention to comparisons with similar clinical studies of propolis.
Thank you for your valuable comment. To the best of our knowledge, based on an extensive search conducted in major scientific databases over the past four years, the conception of this pilot trial as part of my doctoral project, this is the first clinical trial to investigate the use of propolis in patients with stable coronary artery disease and angina.
I appreciate the links you provided. Regarding the study titled “Efficacy and Safety Study of Total Flavonoids of Propolis Dropping Pill to Treat Angina Pectoris”, although it is registered as a clinical trial, we were unable to locate any published results. As for the study “The Effect of Propolis Mouthwash Compared to Chlorhexidine Mouthwash on Oral and Cardiovascular Health”, it investigates the antibacterial properties of propolis and is not directly related to the focus of our study.
The recently published review article from March 2024, “Propolis in the Management of Cardiovascular Disease”, does provide an overview of ongoing research in this area. However, as shown in Table 3 of that review, there are currently no published clinical trials evaluating propolis in the specific context of coronary artery disease similar to our study, which limits the possibilities for direct comparison in the discussion section.
If there are specific parts of the discussion that were difficult to understand, I would be grateful for further clarification so that I can revise and improve the text accordingly.

Reviewer 2 Report
Comments and Suggestions for Authors
The manuscript is well-organized and articulate. The authors candidly present their negative findings and engage in a nuanced discussion concerning the implications of these results. Nevertheless, certain aspects could be enhanced, particularly in relation to the study's rationale, acknowledgment of limitations, and the interpretation of clinical outcomes.
The choice of 200 mg twice daily for six weeks lacks pharmacodynamic justification or reference to prior dose-ranging studies.
Given the heterogeneity in inflammatory responses, a subgroup analysis (e.g., patients with high baseline hs-CRP) might have offered additional insight.
It would be interesting to provide the phytochemical composition of the propolis to enable future comparisons and even driven other studies.
Could certain subgroups (e.g., high hs-CRP, diabetic patients) respond differently to propolis treatment?
How does the anti-inflammatory profile of propolis compare to colchicine or other agents tested in CAD populations?
Were there any changes in secondary lipid, glucose, or endothelial biomarkers that might suggest a trend?
Would imaging (e.g., flow-mediated dilation or coronary CT angiography) offer more sensitive endpoints for future trials?
The authors may consider incorporating a concise section that delineates prospective avenues for future research, particularly those that align with the methodological framework presented in this study.
Author Response
Response to Reviewer:
Thank you for your thoughtful and pertinent comments. I will address each of your points below:
"The manuscript is well-organized and articulate. The authors candidly present their negative findings and engage in a nuanced discussion concerning the implications of these results. Nevertheless, certain aspects could be enhanced, particularly in relation to the study's rationale, acknowledgment of limitations, and the interpretation of clinical outcomes."
I appreciate the reviewer’s observations and will attempt to address the concerns raised, which were highly relevant and insightful.
"The choice of 200 mg twice daily for six weeks lacks pharmacodynamic justification or reference to prior dose-ranging studies."
The choice of 200 mg twice daily was based on prior clinical experience with propolis in other inflammatory conditions in humans, such as in chronic kidney disease undergoing dialysis, COVID-19, and in a meta-analysis showing its association with reductions in C-reactive protein(1–3). In those studies, doses ranged from 200 mg to 1500 mg per day. The 400 mg/day dose was selected for this trial as it had demonstrated safety and good tolerability in previous work.
"Given the heterogeneity in inflammatory responses, a subgroup analysis (e.g., patients with high baseline hs-CRP) might have offered additional insight."
We did conduct this analysis post hoc; however, due to the small sample size and inherent limitations of subgroup analyses, we chose not to include it in the main text. Among the 59 patients enrolled, 32% (n = 19) had a baseline hs-CRP > 2 mg/L. This may raise the hypothesis that propolis did not show positive results in our overall sample due to a considerable proportion of patients not presenting with relevant systemic inflammation. In the CANTOS trial(4), patients with elevated baseline hs-CRP who reached levels <2 mg/L after treatment experienced greater reductions in primary outcomes. Similarly, the recent CLEAR-SYNERGY trial did not demonstrate benefit with colchicine after acute coronary syndrome, and one explanation proposed was that fewer patients achieved hs-CRP reduction below this threshold, potentially influencing the absence of clinical effects.
After your comment, we added a new paragrapher about this issue in the discussion section.
"It would be interesting to provide the phytochemical composition of the propolis to enable future comparisons and even driven other studies."
We have included this information in the discussion section, as also suggested by another reviewer.
"Could certain subgroups (e.g., high hs-CRP, diabetic patients) respond differently to propolis treatment?"
We conducted a subgroup analysis of patients with elevated baseline hs-CRP; however, no significant difference was observed between the propolis and placebo groups after six weeks of treatment (p = 0.75).
"How does the anti-inflammatory profile of propolis compare to colchicine or other agents tested in CAD populations?"
"Were there any changes in secondary lipid, glucose, or endothelial biomarkers that might suggest a trend?"
We did not assess other biochemical or metabolic parameters due to the limited resources available, as the study was conducted in a public hospital setting without any funding.
"Would imaging (e.g., flow-mediated dilation or coronary CT angiography) offer more sensitive endpoints for future trials?"
We believe that a key point for future studies on propolis in coronary artery disease is to recruit a larger sample and to include elevated hs-CRP (>2 mg/L) at baseline as an inclusion criterion. This may help better select a population more likely to respond to anti-inflammatory interventions.
"The authors may consider incorporating a concise section that delineates prospective avenues for future research, particularly those that align with the methodological framework presented in this study."
Thank you for this suggestion.
References:
- Silveira MAD, De Jong D, Berretta AA, Galvão EB dos S, Ribeiro JC, Cerqueira-Silva T, et al. Efficacy of Brazilian green propolis (EPP-AF®) as an adjunct treatment for hospitalized COVID-19 patients: A randomized, controlled clinical trial. Biomedicine and Pharmacotherapy. 2021 Jun 1;138.
- Duarte Silveira MA, Malta-Santos H, Rebouças-Silva J, Teles F, Batista Dos Santos Galvão E, Pinto De Souza S, et al. Effects of Standardized Brazilian Green Propolis Extract (EPP-AF®) on Inflammation in Haemodialysis Patients: A Clinical Trial. Int J Nephrol. 2022;2022.
- Jalali M, Ranjbar T, Mosallanezhad Z, Mahmoodi M, Moosavian SP, Ferns GA, et al. Effect of Propolis Intake on Serum C-Reactive Protein (CRP) and Tumor Necrosis Factor-alpha (TNF-α) Levels in Adults: A Systematic Review and Meta-Analysis of Clinical Trials. Vol. 50, Complementary Therapies in Medicine. Churchill Livingstone; 2020.
- Ridker PM, Everett BM, Thuren T, MacFadyen JG, Chang WH, Ballantyne C, et al. Antiinflammatory Therapy with Canakinumab for Atherosclerotic Disease. New England Journal of Medicine. 2017 Sep 21;377(12):1119–31.

Round 2
Reviewer 1 Report
Comments and Suggestions for Authors
Main notes:
- After looking at the iThenticate report, I suppose that the percent match (28%) is not appropriate. Therefore, it should be corrected.
- The Introduction section is very short and should be expanded by reviewing current scientific publications. For instance, this link could be useful –
- https://pubmed.ncbi.nlm.nih.gov/?term=Coronary+heart+Disease+and+propolis&sort=date&size=50
- The authors should add a separate paragraph to the discussion about the chemical composition of propolis and its bioactive components in more detail, which could exert the identified therapeutic effects.
- Research design should be more clearly described.
- The Discussion section should be corrected for ease of understanding. The authors should also pay more attention to comparisons with similar clinical studies of propolis –
https://clinicaltrials.gov/search?cond=Coronary%20Disease&term=propolis
https://clinicaltrials.gov/search?cond=cardiovascular%20health&term=propolis
Author Response
Author's Reply to the Review Report (Reviewer 1) – round 2.
Research design should be more clearly described. If the authors even made
changes to this, they did not highlight them in any way, so I cannot check
the text "line by line", but please mark the changes made.
Thank you for your comment. All changes made during the first round of revisions, including those related to the description of the research design, were clearly marked in red in the revised manuscript to facilitate your review.
To ensure the methodology section meets the expected standards, we kindly ask if you could specify which particular aspects you believe still require clarification or improvement. As previously mentioned, we have included detailed information on the study design, recruitment process, inclusion and exclusion criteria, endpoints, statistical analysis, and adherence to CONSORT guidelines, with a participant flowchart in Figure 1 and the CONSORT checklist in the supplementary material.
Your specific suggestions would be greatly appreciated to help us address your concerns more precisely and improve the manuscript further.
- After looking at the iThenticate report, I suppose that the percent match (28%) is not appropriate. Therefore, it should be corrected.
Thank you for your comment regarding the iThenticate report. We would like to clarify that the manuscript is entirely original and was developed based on our own research. As is common in academic writing, we used references from previously published studies to support our rationale and to discuss our findings in the context of existing literature.
We are committed to maintaining the integrity of scientific writing and would greatly appreciate it if you could kindly indicate which sections of the manuscript, you believe require attention, based on the similarity report. This will help us better understand your concerns and make the necessary adjustments, if applicable.
- The Introduction section is very short and should be expanded by reviewing current scientific publications.
Thank you for your comments regarding the introduction. I have taken your observations into consideration and added further information to enrich the text and provide a more comprehensive context. These additions can be found in the revised manuscript on lines 49–54 and 58–60.
- The authors should add a separate paragraph to the discussion about the chemical composition of propolis and its bioactive components, which could exert the identified therapeutic effects.
Thank you for the suggestion. I have added additional information regarding the chemical composition of propolis to enhance the contextual background of the manuscript. These additions can be found in the revised manuscript on lines 198-203.
- Research design should be more clearly described.
Thank you for your continued feedback. The methodology section was developed based on the CONSORT guidelines and includes a detailed description of the study design, recruitment process, inclusion and exclusion criteria, interventions, outcomes, and statistical analysis. Additionally, a participant flow diagram is presented in Figure 1 to illustrate the screening, randomization, follow-up, and analysis phases.
To ensure transparency and adherence to reporting standards, we also provided the CONSORT checklist in the supplementary material.
However, as we understand the importance of clarity, we kindly ask if you could indicate which specific aspects of the study design remain unclear or insufficiently described, so we can address them appropriately in our next revision. We remain fully committed to improving the manuscript and greatly value your guidance.
The Discussion section should be corrected for ease of understanding. The authors should also pay more attention to comparisons with similar clinical studies of propolis.
Thank you once again for your feedback. As previously mentioned, to the best of our knowledge and based on an extensive search across major scientific databases, our study appears to be the first published clinical trial investigating the effects of propolis in patients with stable coronary artery disease and angina.
We reviewed the links you kindly provided. However, the trial titled “Efficacy and Safety Study of Total Flavonoids of Propolis Dropping Pill to Treat Angina Pectoris” does not appear to have any published results to date. The study “The Effect of Propolis Mouthwash Compared to Chlorhexidine Mouthwash on Oral and Cardiovascular Health” focuses on antibacterial effects and is not directly comparable to our clinical context.
In light of feedback from other reviewers, we enriched the Discussion section to better address the novelty and implications of our findings, as seen in lines 231–240.
If there are particular aspects of the Discussion that you found unclear or areas where you believe additional comparison or interpretation would be valuable, we would be grateful for your suggestions so that we can improve the manuscript accordingly.
Reviewer 2 Report
Comments and Suggestions for Authors
Following this round of revisions, the document has demonstrated significant enhancement. The authors have duly addressed my earlier feedback, resolving the issues I raised in their manuscript. At this juncture, I am supportive of its publication.
Author Response
We sincerely thank the reviewer for the thoughtful feedback and for acknowledging the improvements made to the manuscript. We greatly appreciate the time and expertise dedicated to reviewing our work, and we are pleased that the revisions have addressed the concerns raised. Thank you for your support of our submission.
Round 3
Reviewer 1 Report
Comments and Suggestions for Authors
The authors have significantly improved this manuscript and can now be accepted for publication.